# A thorough reproduction and evaluation of $\mu$P

**Georgios Vlassis**                                                *gvlassis@ethz.ch*
*D-ITET*
*ETH Zurich*

**Volodymyr Fomichov**                               *volodymyr.fomichov@unidistance.ch*
*Faculty of Mathematics and Computer Science*
*UniDistance*

**David Belius**                                          *david.belius@unidistance.ch*
*Faculty of Mathematics and Computer Science*
*UniDistance*

**Reviewed on OpenReview:** *https://openreview.net/forum?id=AFxEdJwQcp*

## Abstract

This paper is an independent empirical reproduction of the claimed benefits of the $\mu$P parametrization proposed in Yang & Hu (2020) and Yang et al. (2021). Under the so-called Standard Parametrization (SP), the weights of neural networks are initialized from the Gaussian distribution with variance scaling as the inverse of "fan-in", with the learning rate being the same for every layer. While this guarantees that (pre)activations are $\mathcal{O}(1)$ at initialization with respect to width, it causes their scale to be width-dependent during training. To address this, Yang & Hu (2020) and Yang et al. (2021) proposed the Maximal Update Parametrization ($\mu$P), which is also claimed to make the optimal value of various hyperparameters independent of width. However, despite its alleged benefits, $\mu$P has not gained much traction among practitioners. Possibly, this could stem from a lack of thorough independent evaluation of $\mu$P against SP. We address this by independently reproducing the empirical claims of the original works. At the same time, we substantially increase the scale of the experiments, by training 16000 neural networks of sizes from 500 to 1B parameters, and empirically investigate $\mu$P's effect on outputs, gradient updates, weights, training loss and validation loss. We find that generally $\mu$P indeed delivers on its promises, even though this does not always translate to improved generalization.

## 1 Introduction

### 1.1 Related works

Deep learning researchers and practitioners have long understood the importance of initialization and its relation to width. The work LeCun et al. (2002) advocated that weights be sampled from a distribution with mean zero and standard deviation $\frac{1}{\sqrt{\text{fan-in}}}$ (LeCun initialization). Glorot & Bengio (2010) shed further light on why this is helpful, and Sutskever et al. (2013) showed that initialization schemes like this can synergize with momentum methods.

Subsequent theoretical work (Jacot et al., 2018; Chizat et al., 2019) has demonstrated that using neural networks with these initialization schemes leads to so called "lazy training" in the infinite-width limit. Specifically, in that limit they converge to kernel machines. These limiting kernels do not depend on the dataset at all (that is, they are not learned). This means that in the infinite-width limit, feature learning becomes impossible, possibly hurting performance.

Motivated by the desire to escape this "lazy training trap", Yang & Hu (2020) introduced a parametrization[1] that is supposed to maintain the ability to learn features at infinite-width. Starting from the desideratum that (pre)activations be $\mathcal{O}(1)$ throughout training, and using the theory developed in the Tensor Programs (TP) series of papers (Yang, 2019a;b; 2020a; Yang & Littwin, 2021; Yang, 2020b; Yang & Hu, 2020; Littwin & Yang, 2022; Yang et al., 2023a; 2021; 2023b; Yang & Hu, 2020), they arrive at the $\mu$P parametrization. For many hyperparameters, $\mu$P is also claimed to stabilize optimal values as width varies, a property that is exploited in Yang et al. (2021) for hyperparameter optimization. In this paradigm, called $\mu$Transfer, optimal hyperparameters are discovered cheaply for a small, proxy network, and then zero-shot transferred to a big, target network. Lastly, Yang et al. (2021) claim that wider networks should in general have lower training loss under $\mu$P, which is not necessarily true under SP.

Since its proposal, $\mu$P has been used in a limited number of published works. For the case of Large Language Models (LLMs), it has been used by Dey et al. (2023a) (Cerebras-GPT), Li et al. (2023) (FLM-101B), Dey et al. (2023b) (BTLM-3B-8K), Liu et al. (2023) (CrystalCoder), Hu et al. (2024) (MiniCPM) and Li et al. (2024) (Tele-FLM). Intriguingly Achiam et al. (2023) (GPT-4) includes Yang & Littwin (2021) in the references without explicitly citing it, leaving it unclear if it uses $\mu$P or not. Outside of the LLM world, $\mu$P was used in Cabannes et al. (2023) to ensure that a fixed learning rate was reasonable for every width they tested, and in Beaini et al. (2023), which included $\mu$P in their GNN library Graphium targeted at molecular learning.

## 1.2 Objectives

The above works using $\mu$P nearly always assume its benefits, taking at face value that $\mu$P is preferable over SP, without ablating with respect to the parametrization. Besides the original papers (Yang & Hu, 2020; Yang et al., 2021), the only work that investigates the claimed advantages of $\mu$P over SP is Lingle (2024). It studies whether $\mu$P indeed stabilizes the optimal learning rate for many architectural variations of a transformer, and finds that it does for most but not all of these variations.

In this paper, we will thoroughly investigate the alleged benefits of $\mu$P and compare it head-to-head with SP. We expand the scale of the existing $\mu$P versus SP comparisons (Yang & Hu, 2020; Yang et al., 2021; Lingle, 2024), by including additional architectures and domains, scaling to narrower and wider networks, performing a denser hyperparameter sweep, training for more random seeds and training for longer. In total, we train 15760 networks, ranging from 500 to 1B parameters. Our ultimate goal is to understand whether and to what extent the promises of $\mu$P hold in practice, and if and when it should be preferred over SP.

Our work is fundamentally an independent reproduction of Yang & Hu (2020) and Yang et al. (2021). Hence, we made every effort that our results are reproducible themselves. The complete repository of the training code[2] is already available online.

## 1.3 Findings

Our findings can be summarized as follows:

1. Inspecting the norm of coordinate-wise network outputs reveals that they indeed are $\mathcal{O}(1)$ under $\mu$P, while heavily depending on width under SP. Similarly, the norms of coordinate-wise gradient updates and weights decay exponentially with width under both SP and $\mu$P.

2. For $\mu$P, and unlike SP, the best (with respect to the training loss) learning rate indeed stays *approximately* constant as width increases. Thus, $\mu$Transfer, in contrast to "naive" hyperparameter tuning with SP, indeed enables zero-shot hyperparameter transfer, from narrow (and thus cheaply trainable) networks to wider ones.

3. Under $\mu$P, wider networks *in general* outperform (in training loss) narrower networks. Under SP this trend is much less visible, although sometimes present.

---

[1] A parametrization is simply a prescription according to which the weights of a network are initialized and updated.
[2] https://github.com/gvlassis/ant

4. Points 2 and 3 do not always translate to better generalization. That is, the optimal $\mu$P network often has worse validation loss than the optimal SP network.

5. With SP, we observed some wide networks diverging. Specifically, the wider the network, the more likely it was to diverge. In contrast, almost none of the networks diverged with $\mu$P.

6. The benefits of $\mu$P seem to be more evident for transformers.

In summary, we found that $\mu$P *mostly* performs as expected in terms of stability of optimal learning rates, but does often leads to worse generalization.

## 2 $\mu$P summary

We start with a high-level summary of the Tensor Programs framework (Yang, 2019a).

In this framework, the initial weights and learning rates of a neural network are scaled in terms of a parameter matrix's "fan-in" and "fan-out". Their precise meaning for different types of layers are as follows:

1. The parameter matrix of biases of a linear layer has fan-in $= 1$, and fan-out equal to the activation dimension.

2. The parameters of a convolutional layer are viewed as a collection of input_channels $\times$ output_channels matrices (where fan-in = input_channels and fan-out = output_channels). The number of such matrices for a layer is kernel_width $\times$ kernel_height.

3. Biases and weights of layer normalization layers are treated the same as biases of linear layers.

4. The class embedding of Vision Transformers (ViTs) has fan-in $= 1$ and fan-out $= d$, where $d$ is the model dimension.

5. The embedding operation of a transformer is viewed as a matrix multiplication between the embedding table and a one-hot vector representing a token of the vocabulary. Therefore, the embedding table has fan-in = vocabulary_size and fan-out $= d$.

With these conventions, assume $\theta \in \mathbb{R}^{\text{fan-in} \times \text{fan-out}}$ is a parameter matrix of a neural network.

Under SP, the initialization and update rules are:

$$\theta_0 \sim \begin{cases} \mathcal{N}(\mu, c^2) & \text{if fan-in} = 1, \\ \mathcal{N}(0, c^2 \cdot \frac{1}{\text{fan-in}}) & \text{if fan-in} > 1, \end{cases} \tag{1}$$

$$\theta_{t+1} \leftarrow \theta_t - k \cdot f(\nabla \theta_t), \tag{2}$$

for a function $f$, where $\mu$, $c$ and $k$ are hyperparameters that do not scale with width. Note that $c = 0$ is possible (e.g. biases are often initialized to zero). Different choices of $f$ lead to different optimizers (e.g. for $f = \text{id}$ we recover SGD, while another choice leads to Adam).

Under the $\mu$P, the initialization and update rules are instead:

$$\theta_0 \sim \begin{cases} \mathcal{N}(\mu, c^2) & \text{if fan-in} = 1, \\ \mathcal{N}(0, c^2 \cdot s^2) & \text{if fan-in} > 1, \end{cases} \tag{3}$$

$$\theta_{t+1} \leftarrow \theta_t - k \cdot \gamma \cdot f(\nabla \theta_t), \tag{4}$$

where $s$ and $\gamma$ are scaled with width as specified in Table 1. In addition, the scale in a self-attention layer of dimension $d$ should be changed from $\frac{1}{\sqrt{d}}$ to $\frac{1}{d}$.

The constants $\mu$, $c$ and $k$ do not have to match between SP and $\mu$P. Moreover, they can be chosen arbitrarily for every parameter matrix of the network. This allows us to make SP and $\mu$P *exactly* equivalent for a base width. We can do so by inserting width-independent constants in front of the $\mu$, $c$ and $k$ of $\mu$P. The constants to be inserted are obtained from equating the initializations (equation 1 and equation 3) and the update rules (equation 2 and equation 4).

Table 1: Standard deviation and learning rate scaling in $\mu$P

|  | fan-out $\to \infty$ | fan-in, fan-out $\to \infty$ | fan-in $\to \infty$ |
|---|---|---|---|
| $s$ | $1/\sqrt{\text{fan-in}}$ | $1/\sqrt{\text{fan-in}}$ | $1/\text{fan-in}$ |
| $\gamma$ (SGD) | fan-out | $1$ | $1/\text{fan-in}$ |
| $\gamma$ (Adam) | $1$ | $1/\text{fan-in}$ | $1/\text{fan-in}$ |

## 3 Experimental setup

We experimented with four architectures, across five datasets. Specifically, we tested a 3-layer MLP on the California Housing and the MNIST datasets, a VGG11 CNN and a ViT on CIFAR-10, and a transformer on Tiny Shakespeare and WikiText-103.

For every architecture we chose a base width, and then trained networks of widths $\zeta \times$ base_width while varying $\zeta$. We ran comprehensive experiments for each architecture and dataset combination. For every combination, we picked multipliers to make SP and $\mu$P exactly equivalent for the base width $\zeta = 1$ (as described in the previous section). We swept the learning rate hyperparameter $k$, training 16 networks for every value.

For our MLP on California Housing, we followed Yang et al. (2021, Figure 5) in plotting the norm of coordinate-wise outputs to test $\mu$P's stabilizing effect on them. We also did the same for weights and gradient updates.

In all settings, we compared performance for different hyperparameter values at varying width, producing curves like those of Yang et al. (2021, Figure 1). Specifically, we collected the minimum training and validation losses, and plotted their mean, along with one standard deviation error bars for both parametrizations.

For all the experiments, we set the initialization scale $c$ to $1/10$ and used the Adam optimizer (Kingma & Ba, 2017) with PyTorch's defaults. Additionally, we trained without weight-decay or data augmentation.

In total, we trained 15760 neural networks, spanning from 500 to 1B parameters, which needed 3200 hours when using an NVIDIA A100 [3].

### 3.1 MLP on California Housing

The California Housing dataset (Pace & Barry, 1997) is a tabular regression dataset with the goal of predicting the median house value for a geographical block in California from eight real-valued features. It consists of 20640 samples, out of which we held out 2000 for validation and 2000 for testing.

We used an MLP with two hidden layers, and gave them a base width of 16. We trained networks corresponding to width multipliers from $\zeta = 1$ (width = 16, parameters = 433) to $\zeta = 512$ (width = 8192, parameters = 67M). For each width we trained with 16 different learning rate multipliers $k$, geometrically spaced between $10^{-5}$ and 1. Each training run consisted of 50000 mini-batches of size 16.

Overall, we trained 5120 MLPs, which took around 200 hours on an NVIDIA A100.

### 3.2 MLP on MNIST

The MNIST dataset (LeCun et al., 1998) is an image classification dataset with ten classes, one for each handwritten digit. It contains 70000 greyscale images, of size $28 \times 28$. We held out 10000 images for validation and 10000 for testing.

The images were flattened to 784-dimensional vectors and passed to MLPs with two hidden layers with base width 16. Similarly to in Section 3.1, we used an MLP with two hidden layers and base width 16. We varied $\zeta$ from $\zeta = 1$ (width = 16, parameters = 13K) to $\zeta = 256$ (width = 4096, parameters = 20M). The remaining training details are the same as in Section 3.1, with the difference that here we used 20000 mini-batches.

---

[3]At the time of publication, this would cost on the order of 13000$ on AWS (Amazon, 2024).

Overall, we trained 4608 MLPs, which needed about 100 GPU hours.

### 3.3  VGG11 on CIFAR-10

The CIFAR-10 dataset (Krizhevsky et al., 2009) is an image classification dataset where one tries to classify an image in one of ten classes. There are 60000 images, of size $3 \times 32 \times 32$. We held out 10000 images for validation and 10000 for testing.

We used the VGG11 architecture (Simonyan & Zisserman, 2014) with four convolutional stages. The stages had base width[4] 4, 8, 16 and 32 respectively. The classifier head had base width 20, and 0.5 dropout probability. We tested networks from $\zeta = 1$ (max_channels = 32, parameters = 21K) to $\zeta = 128$ (max_channels = 4096, parameters = 336M). We tried eight geometrically spaced values for the learning rate multiplier $k$, between $6 \cdot 10^{-5}$ and 0.01. Each training run consisted of 50000 mini-batches, of size 32.

In aggregate, we trained 2048 CNNs, for about 500 GPU hours.

### 3.4  ViT on CIFAR-10

We used the ViT architecture (Dosovitskiy et al., 2020) with a patch size of four and six blocks of base width 32, eight heads, expansion factor of one and 0.1 dropout probability. For positional embeddings we used sinusoidal positional encodings. We tested networks from $\zeta = 1$ (width = 32, parameters = 34K) to $\zeta = 128$ (width = 4096, parameters = 504M). The remaining training details follow Section 3.3.

In total, we trained 2048 ViTs, which took 1000 GPU hours.

### 3.5  Transformer on Tiny Shakespeare

The Tiny Shakespeare dataset (Karpathy, 2015) is a subset of Shakespeare's works in a single 40000 lines file. Language models trained from scratch on this dataset can produce samples that look very close to the original. We tokenized the dataset with the GPT-2 (Radford et al., 2019) tokenizer, leading to 300K tokens. We held out 25K tokens for validation and 25K tokens for testing.

We used the transformer architecture Vaswani et al. (2017) with a context of 128 tokens and six blocks of base width 32, eight heads, expansion factor of four and no dropout. For positional embeddings we used sinusoidal positional encodings. We tested networks from $\zeta = 1$ (width = 32, parameters = 3.3M) to $\zeta = 32$ (width = 1024, parameters = 180M). We tried eight geometrically spaced values for the learning rate multiplier $k$, between $6 \cdot 10^{-4}$ and 0.1. Each training run consisted of 20000 mini-batches of size 32.

Overall, we trained 1536 transformers, for approximately 300 hours.

### 3.6  Transformer on WikiText-103

The WikiText-103 dataset (Merity et al., 2016) is a collection of 1.8M verified Good and Featured articles from Wikipedia. We held out 4K articles for validation and 4.5K for testing. We tokenized the dataset with the GPT-2 (Radford et al., 2019) tokenizer, leading to 114M tokens.

The architecture used was a scaled-up version of Section 3.5, with the context doubled to 256 tokens, twelve transformer blocks with base width 144 and twelve attention heads. We varied $\zeta$ from $\zeta = 1$ (width = 144, parameters = 17M) to $\zeta = 16$ (width = 2304, parameters = 997M). We tried ten geometrically spaced values for the learning rate multiplier $k$, between $10^{-5}$ and 0.1. Each training run consisted of 8000 mini-batches, of size 512 (1B tokens).

Because each training run is *much* more computationally expensive than in the previous settings, we trained four networks (instead of 16) for every value of $k$. Moreover, as is the norm in such settings, we used $\beta_2 = 0.95$ for Adam.

---

[4]The width of a convolutional layer is simply the number of its output channels.

Overall, we trained 400 transformers, in 1100 hours.

## 4  Main results

### 4.1  MLP on California Housing

#### 4.1.1  Scale of activations

As our first experiment, we measured the average coordinate-wise norm of the output of our MLP architecture on California Housing, described in Section 3.1. We did this for width multipliers from $\zeta = 1$ to $\zeta = 512$ and for twelve batches. We then compared SP with $\mu$P to see the impact of parametrization. According to theory, outputs should be width-dependent under SP, and width-independent under $\mu$P. The results are presented in Figure 1. For SP, we can see that the scale of the outputs rapidly increases as we increase the width. On the contrary, for $\mu$P, the norm is stable with respect to the width. The results are as expected, and mirror Yang et al. (2021, Figure 5).

We did the same for the gradient updates and the weight norms of the last hidden layer of our MLP. According to theoretical claims in Yang & Hu (2020), the coordinate-wise norms of the gradient updates and the weights, under both SP and $\mu$P, decay exponentially with width. Again in Figure 1, we confirm these claims. Interestingly, the gradient update curves are more stable under $\mu$P, with small spikes appearing for nearly all batches under SP.

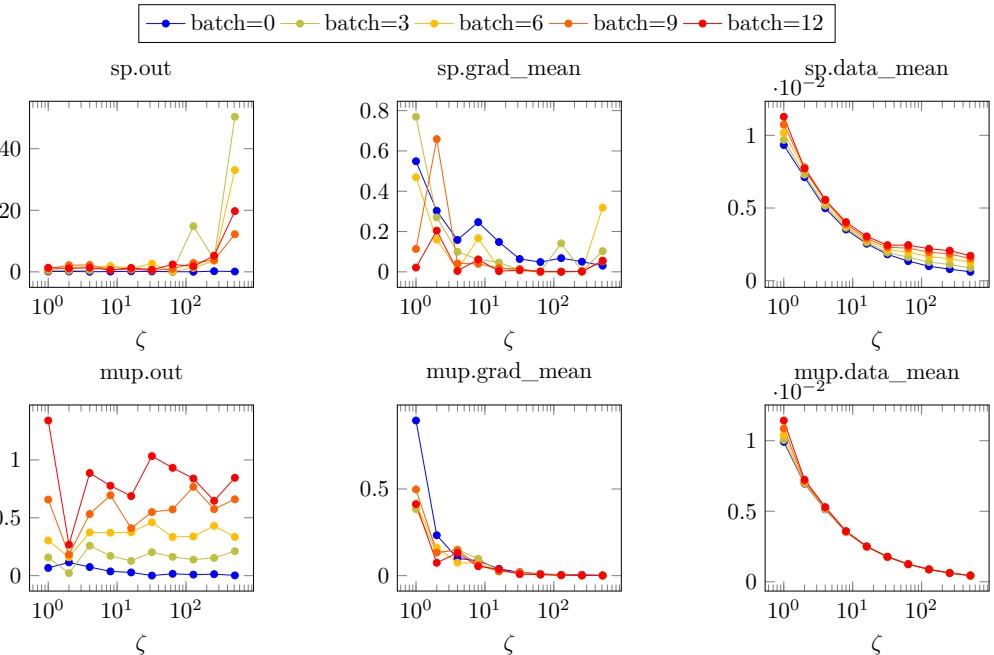

Figure 1: Scale of network outputs (left), gradient updates (middle) and weights (right) as function of the width multiplier $\zeta$. $\mu$P stabilizes the scale of the outputs with respect to width.

#### 4.1.2  Train and validation loss

The results for the stability of the hyperparameters with changing width are shown in Figure 2. We observe that the training loss curves for both SP and $\mu$P are quite noisy, with the error bars for different widths overlapping. This indicates that in some cases the benefits of $\mu$P are detectable only when averaging over many training runs, highlighting the need for many random seeds.

Under SP, the optimal learning rate multiplier $k$ with respect to the training loss shifts around an order of magnitude to the left as the width increases. On the other hand, it stays approximately constant under $\mu$P.

Moreover, under $\mu$P, the curves are somewhat flatter, which means that the networks are less sensitive to the exact value of $k$.

For SP, wider networks do not consistently outperform narrower ones in terms of training loss, except for a small range of low values of $k$, and the difference is slight. Meanwhile, this trend is much stronger for $\mu$P, and observed for a wider range of $k$. The validation loss curves show similar behavior, but are less noisy.

Comparing best performing networks with respect to the training loss, we see that the best SP network has $\zeta = 128$, $k = 10^{-4}$ and min_training_loss $= 6.52 \cdot 10^{-2}$ [5] ($R^2 = 0.66$), while the best $\mu$P network has $\zeta = 512$, $k = 3 \cdot 10^{-4}$ and min_training_loss $= 6.78 \cdot 10^{-2}$ ($R^2 = 0.62$). Thus, for SP the third widest network performs the best, while, consistently with theory, the widest network performs the best for $\mu$P. With respect to the validation loss, the best networks have $\zeta = 8$, $k = 6 \cdot 10^{-4}$ and min_val_loss $= 0.48$ ($R^2 = 0.55$) for SP and $\zeta = 512$, $k = 0.1$ and min_val_loss $= 0.47$ ($R^2 = 0.54$) for $\mu$P. Hence, $\mu$P has a worse best performing network in terms of training loss in comparison to SP, and a slightly better, but comparable, one in terms of the validation loss.

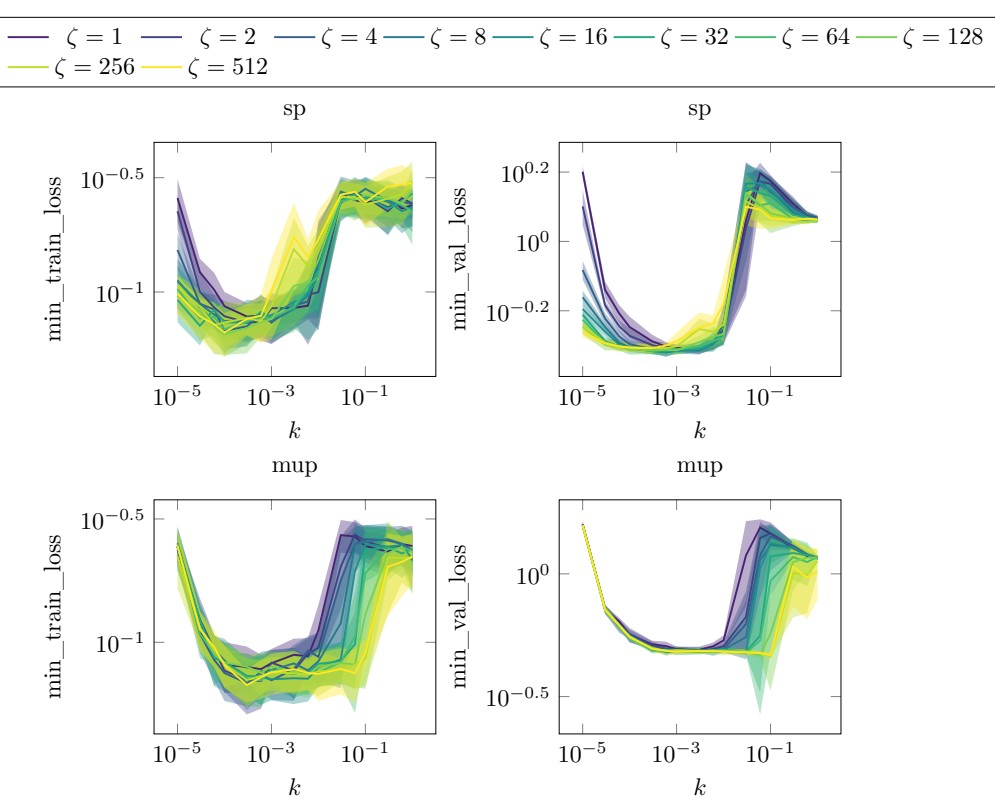

Figure 2: MLP on California Housing. The $y$-axes correspond to MSE loss, and the $x$-axes correspond to the learning rate multiplier $k$. The solid curves show the mean, and the shaded regions $\pm$ one standard deviation. Albeit the plots are quite noisy, we see both that $\mu$P stabilizes the best $k$'s, and that it enables wider networks to more consistently outperform their narrow counterparts.

## 4.2   MLP on MNIST

Training the same architecture as in Section 4.1 on MNIST, we obtained Figure 3.

The noise level it is significantly lower than in Figure 2, though the standard deviation of the train loss is still quite large.

---

[5]Confidence intervals for the minimum training and validation losses for all the settings can be found in Table 2.

The best learning rate multiplier $k$ in terms of training loss shifts around half an order of magnitude to the left for SP as the width increases. Contrary to theory, it shifts about the same, but to the right, for $\mu$P. This was actually the only setting in which we observed the optimum significantly shifting for $\mu$P. We suspect that this is due to the narrower widths being subject to finite size effects (such as the non-vanishing variance of the network output, discussed in Section D.2 of Yang et al. (2021)). However, in terms of validation loss, the optimum $k$ shifts approximately two orders of magnitude for SP, while indeed staying constant for $\mu$P. Even for SP, we see that wider networks tend to outperform their narrower counterparts, both in terms of training and in terms of validation loss (with the exception of $\zeta = 256$). This trend is nevertheless stronger for $\mu$P.

With respect to the training loss, the optimal SP network has $\zeta = 256$, $k = 10^{-4}$ and min_training_loss $= 0$ (acc. $= 99.06\%$) and the optimal $\mu$P network has $\zeta = 256$, $k = 6 \cdot 10^{-3}$ and min_training_loss $= 3 \cdot 10^{-6}$ (acc. $= 98.25\%$). With respect to the validation loss, the best performing SP network has $\zeta = 256$, $k = 3 \cdot 10^{-5}$ and min_val_loss $= 2.94 \cdot 10^{-2}$ (acc. $= 96.88\%$) for SP and $\zeta = 128$, $k = 10^{-3}$ and min_val_loss $= 3.41 \cdot 10^{-2}$ (acc. $= 96.69\%$) for $\mu$P. Therefore, in this setting, SP marginally outperforms $\mu$P in terms of both training (contrary to theory) and validation losses.

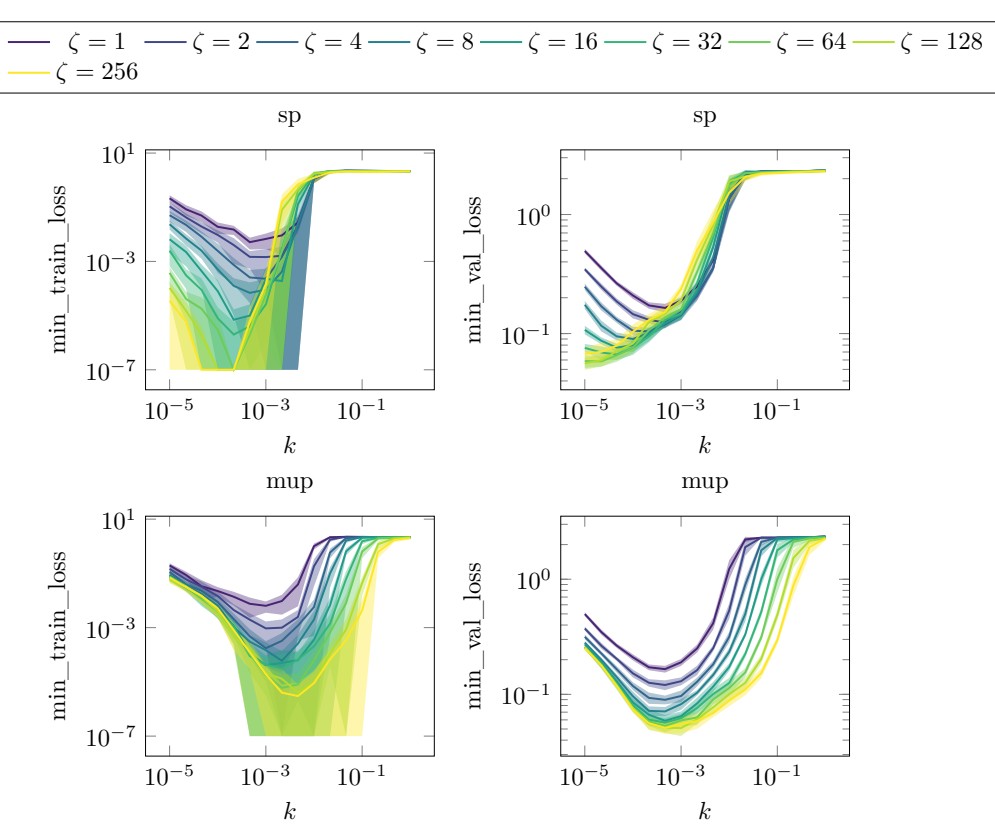

Figure 3: MLP on MNIST. The plots show mean (solid curves) $\pm 1$ std. dev. (shaded regions) of cross entropy loss versus the learning rate multiplier $k$. In this setting $\mu$P does not stabilize the best $k$'s as much as in our other experiments, but it does enable wider networks to more consistently outperform narrow ones. The flat regions are due to clipping values below $10^{-7}$.

## 4.3 VGG11 on CIFAR-10

The results for the VGG11 architecture on CIFAR-10 are shown in Figure 4.

As in Figure 2, the loss curves are quite noisy. Unlike Figure 2, here the noise levels of validation and training loss curves are similar.

Under SP, the best learning rate multiplier $k$ with respect to the training loss shifts around half an order of magnitude to the left as the width increases. On the other hand, it stays roughly constant under $\mu$P. For SP, wider networks consistently outperform narrower ones in terms of training loss only for $k \leq 10^{-4}$. Meanwhile, consistently with theory, for $\mu$P this trend is observed for every $k$. As for the validation loss curves, they are very similar to the ones for the training loss.

Comparing best performing networks with respect to the training loss, we see that the optimal network for SP has $\zeta = 128$, $k = 6 \cdot 10^{-5}$ and min_training_loss $= 2.61 \cdot 10^{-5}$ (acc. $= 99.50\%$), while $\mu$P has $\zeta = 64$, $k = 3 \cdot 10^{-3}$ and min_training_loss $= 1.27 \cdot 10^{-4}$ (acc. $= 99.12\%$). Hence, as in Section 4.2 and in contrast to theoretical predictions, it is actually not the widest network that performs the best for $\mu$P. With respect to the validation loss, the best performing network has $\zeta = 128$, $k = 6 \cdot 10^{-5}$ and min_val_loss $= 0.74$ (acc. $= 77.66\%$) for SP and $\zeta = 128$, $k = 10^{-3}$ and min_val_loss $= 0.76$ (acc. $= 76.94\%$) for $\mu$P. In summary, even though the differences are small, the best performing SP networks outperform the best performing $\mu$P networks in terms of both training and validation loss.

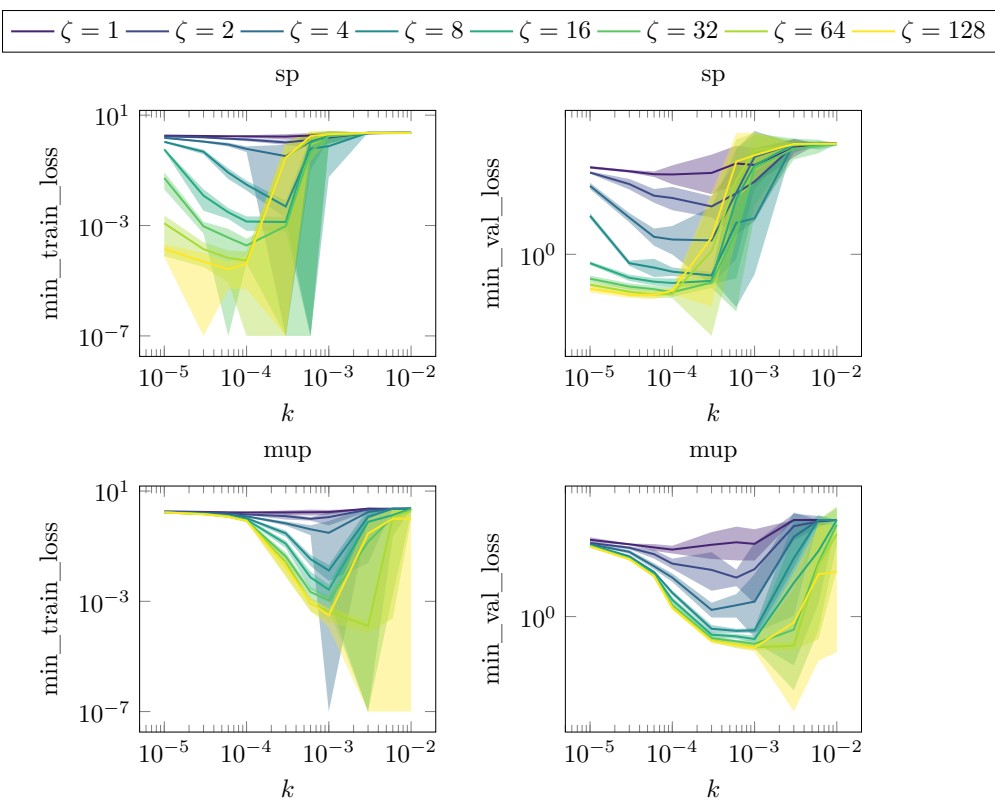

Figure 4: VGG11 on CIFAR-10. We plot cross entropy loss vs learning rate multiplier $k$. Under $\mu$P, the best $k$'s do not shift and wider networks perform better.

## 4.4 ViT on CIFAR-10

The results for the ViT architecture on CIFAR-10 are shown in Figure 5.

The error bars are much tighter than in Figure 2 and Figure 4, both for the training and for the validation loss curves.

Under SP, we see that the best learning rate multiplier $k$ with respect to the training loss shifts around two orders of magnitude to the left as the width increases. On the other hand, it stays almost constant under $\mu$P.

For SP, for some $k$, we can see wider networks outperforming narrower ones in terms of training loss. However, this can only be observed for a small range of learning rates close to the smallest we tried, and the difference is slight. Meanwhile, for $\mu$P the loss is roughly monotonically decreasing in width for a larger range of $k$, centered on the (approximately width-independent) optimum.

Comparing best performing networks with respect to the training loss, the best SP network was the second widest, with $\zeta = 64$, $k = 3 \cdot 10^{-5}$ and min_training_loss = $5.62 \cdot 10^{-3}$ (acc. = 98.20%), while for $\mu$P it was obtained for the third widest, with $\zeta = 32$, $k = 3 \cdot 10^{-3}$ and min_training_loss = 0.01 (acc. = 97.41%). Hence, though the $\mu$P networks exhibit better stability than the SP networks in terms of best learning rate, as well as higher monotonicity of the train loss relative to width, the best SP network in fact slightly outperforms the best $\mu$P in terms of training loss by half an order of magnitude.

The shape of the validation loss curves is qualitatively similar to the training loss curves, with the notable exception of the validation loss curve of the widest $\mu$P network, which has a peculiar peak[6].

Furthermore, for SP and contrary to theory also for $\mu$P wider networks perform worse in terms of validation loss. With respect to the validation loss, the best performing SP network had $\zeta = 4$, $k = 10^{-4}$ and min_val_loss = 1.07 (acc. = 62.56%), and the best performing $\mu$P network had $\zeta = 4$, $k = 3 \cdot 10^{-3}$ and min_val_loss = 1.09 (acc. = 61.87%). Hence, also in terms of validation loss, the best SP network outperforms the best $\mu$P network.

Another interesting observation is that, as seen in Table 2, some SP networks diverged during training. Specifically, one network diverged for $\zeta = 32$, two networks diverged for $\zeta = 64$ and five networks diverged for $\zeta = 128$. By contrast, no $\mu$P networks diverged. The pattern suggests that SP networks become increasingly unstable as we increase the width, while $\mu$P networks are more stable, consistently with the theory behind $\mu$P. Eight networks is a tiny number compared to the 1280 total networks we trained, so this could go unnoticed had the scale of our experiments been smaller. However, this could prove crucial for the training of extremely big networks.

### 4.5 Transformer on Tiny Shakespeare

The results for the transformer on the Tiny Shakespeare dataset are shown in Figure 6.

The training curves are very similar to those reported for a transformer language model in Yang et al. (2021, Figure 1). Moreover, we notice that the training and validation curves have significantly higher standard deviation for SP.

Under SP, we see that the best learning rate multiplier $k$ with respect to the training loss shifts around two orders of magnitude to the left as the width increases. On the other hand, it stays almost constant under $\mu$P. Furthermore, like in Figure 2, under $\mu$P the curves are flatter, meaning that the networks are less sensitive to the value of $k$. For a small range of $k$ wider SP networks outperform narrower ones in terms of training loss, while (as predicted by theory) for $\mu$P this behavior is much more consistent, for almost every $k$.

Quantitatively, the best network for SP was obtained for $\zeta = 32$, $k = 3 \cdot 10^{-4}$ with min_training_loss = 0.17 (perplexity = 1.18), while for $\mu$P it was obtained for $\zeta = 32$, $k = 6 \cdot 10^{-3}$ with min_training_loss = 0.18 (perplexity = 1.20). Hence, in terms of training loss, the best SP network somewhat outperformed the best $\mu$P network.

The validation loss curves are qualitatively similar to their training loss counterparts. For SP, the best network in terms of validation loss was obtained for $\zeta = 16$, $k = 10^{-4}$ with min_val_loss = 96.54 (perplexity = 4.06), while for $\mu$P it was obtained for $\zeta = 32$, $k = 6 \cdot 10^{-3}$ with min_val_loss = 4.72 (perplexity = 112.16). Hence, in terms of validation loss, the best SP network moderately outperformed the best $\mu$P network.

---

[6]We could not clarify the origin of this peak. To rule out a bug in the implementation we carefully reviewed the code and reran the experiment for this width, producing the same results.

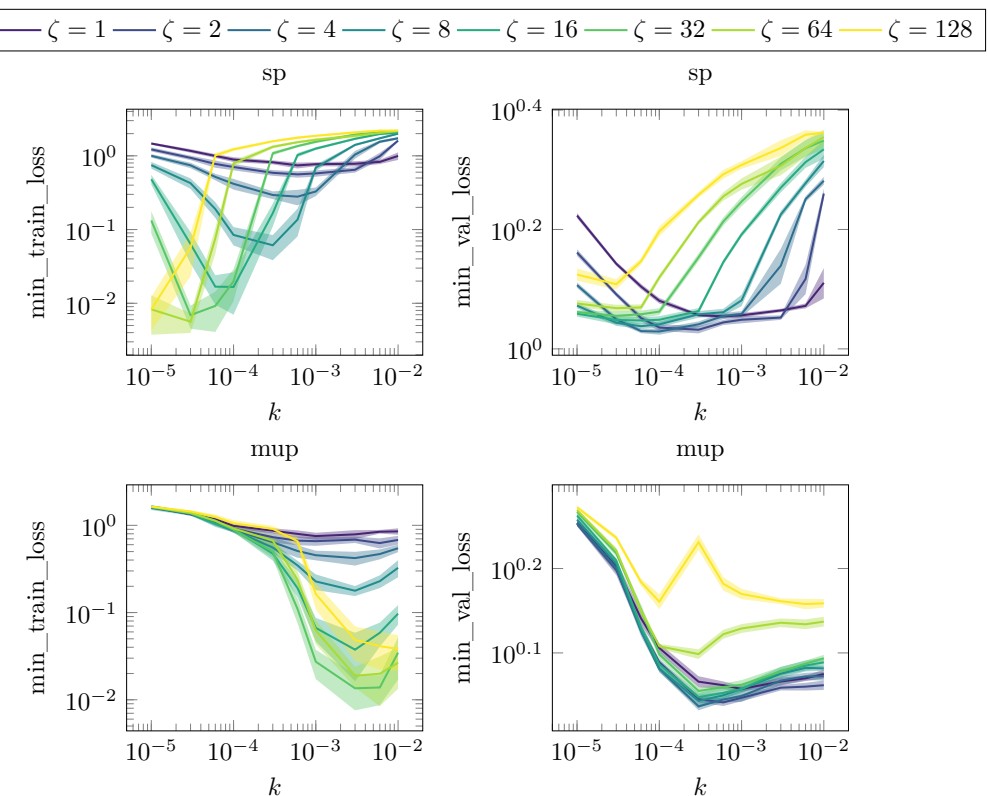

Figure 5: ViT on CIFAR-10. We again plot the mean (solid) $\pm$ 1 std. dev. (shaded) of cross entropy loss vs learning rate multiplier $k$. Broadly speaking, $\mu$P fixes the best $k$'s and leads to wider networks mostly outperforming narrower ones, consistently with theory.

Moreover, we again see in Table 2 that some (precisely 74) networks diverged under SP. Similarly to Section 4.4, we observed that, the wider the network, the more likely it is to diverge. For $\mu$P on the other hand no networks diverged.

### 4.6 Transformer on WikiText-103

Lastly, the results for the transformer on the WikiText-103 dataset are shown in Figure 7.

The loss curves are very similar to those in Section 4.5, but exhibit even less noise. This could be explained by the significantly bigger batch size (512 vs 32), which reduces the stochasticity of the training trajectory.

We see that for SP the best learning multiplier $k$ with respect to the training loss decreases roughly an order of magnitude as we make the architecture wider. In contrast, $\mu$P stabilizes this optimum. Moreover, under $\mu$P the trend of wider networks outperforming narrower ones in terms of training loss is stronger, even though it can also been observed for $k \leq 10^{-3}$ under SP.

In terms of training loss, the best SP network has $\zeta = 16$, $k = 6 \cdot 10^{-4}$ with min_training_loss $= 1.71$ (perplexity $= 5.52$), and the best $\mu$P network has $\zeta = 16$, $k = 6 \cdot 10^{-3}$ with min_training_loss $= 1.40$ (perplexity $= 4.06$). Thus, this is the only setting in which $\mu$P significantly outperformed SP. This also holds for the validation loss, where the best SP network has $\zeta = 8$, $k = 3 \cdot 10^{-4}$ with min_val_loss $= 2.99$ (perplexity $= 19.88$), and the best $\mu$P network has $\zeta = 16$, $k = 6 \cdot 10^{-4}$ with min_val_loss $= 1.40$ (perplexity $= 19.10$).

In Table 2 we see that here some (precisely four) $\mu$P networks also diverged, although their number was still much lower than the SP networks (namely twelve) which diverged.

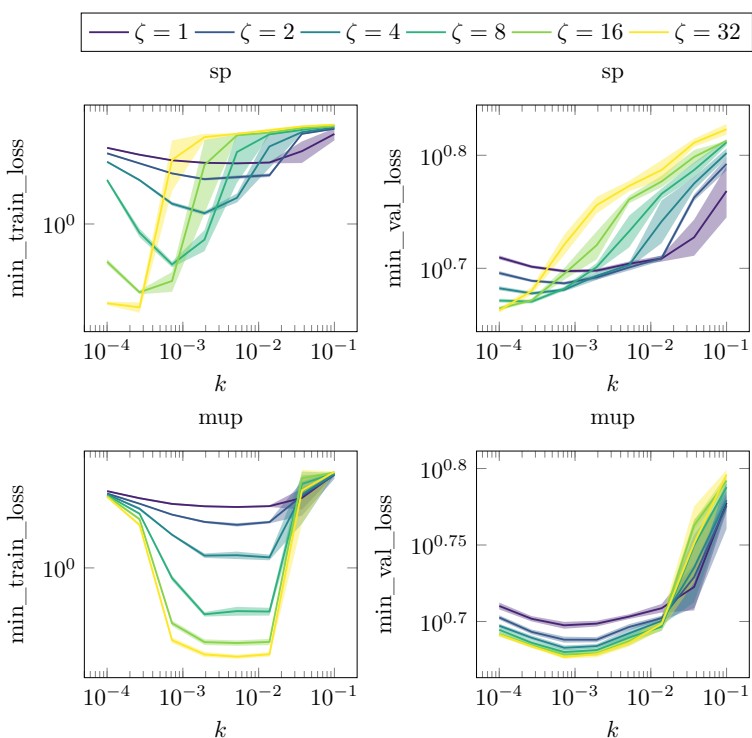

Figure 6: Transformer on Tiny Shakespeare. The $y$-axes show cross entropy loss, and the $x$-axes show the learning rate multiplier $k$. Under SP, the best $k$'s shift significantly, and wider networks do not always lead to better performance. The results for $\mu$P are better in both regards.

## 5  Summary

Our most important empirical results are summarized in Table 2. We see that $\mu$P performed similarly but worse than SP in four of the six settings, significantly worse than SP in one setting, and significantly better than SP in one setting. Thus, the claim that $\mu$P encourages feature learning and therefore better generalization than SP is not consistently supported by our experiments.

Table 2: Summary of our results, showing mean minimum train and validation losses $\pm$ one standard deviation. For the majority of cases, the standard deviation is relatively low, indicating that we can indeed draw conclusions from the mean of the minimum training and validation losses.

| Setting | Parametrization | min_train_loss | min_val_loss | Networks diverged |
|---|---|---|---|---|
| MLP on California Housing | SP | $(\mathbf{6.52 \pm 0.12}) \cdot \mathbf{10^{-2}}$ | $0.48 \pm 0.01$ | **0** |
| | $\mu$P | $(6.78 \pm 0.10) \cdot 10^{-2}$ | $\mathbf{0.47 \pm 0.01}$ | **0** |
| MLP on MNIST | SP | $\mathbf{0.00 \pm 0.00}$ | $(\mathbf{2.94 \pm 0.37}) \cdot \mathbf{10^{-2}}$ | **0** |
| | $\mu$P | $(3 \pm 1) \cdot 10^{-6}$ | $(3.41 \pm 0.36) \cdot 10^{-2}$ | **0** |
| VGG11 on CIFAR-10 | SP | $(\mathbf{2.61 \pm 1.15}) \cdot \mathbf{10^{-5}}$ | $\mathbf{0.74 \pm 0.01}$ | **0** |
| | $\mu$P | $(1.27 \pm 0.25) \cdot 10^{-4}$ | $0.76 \pm 0.02$ | **0** |
| ViT on CIFAR-10 | SP | $(\mathbf{5.62 \pm 1.66}) \cdot \mathbf{10^{-3}}$ | $\mathbf{1.07 \pm 0.01}$ | 8 |
| | $\mu$P | $0.01 \pm 0.00$ | $1.09 \pm 0.01$ | **0** |
| Transformer on Tiny Shakespeare | SP | $\mathbf{0.17 \pm 0.02}$ | $\mathbf{4.57 \pm 0.02}$ | 74 |
| | $\mu$P | $0.18 \pm 0.02$ | $4.72 \pm 0.01$ | **0** |
| Transformer on WikiText-103 | SP | $1.71 \pm 0.03$ | $2.99 \pm 0.01$ | 12 |
| | $\mu$P | $\mathbf{1.40 \pm 0.06}$ | $\mathbf{2.95 \pm 0.01}$ | **4** |

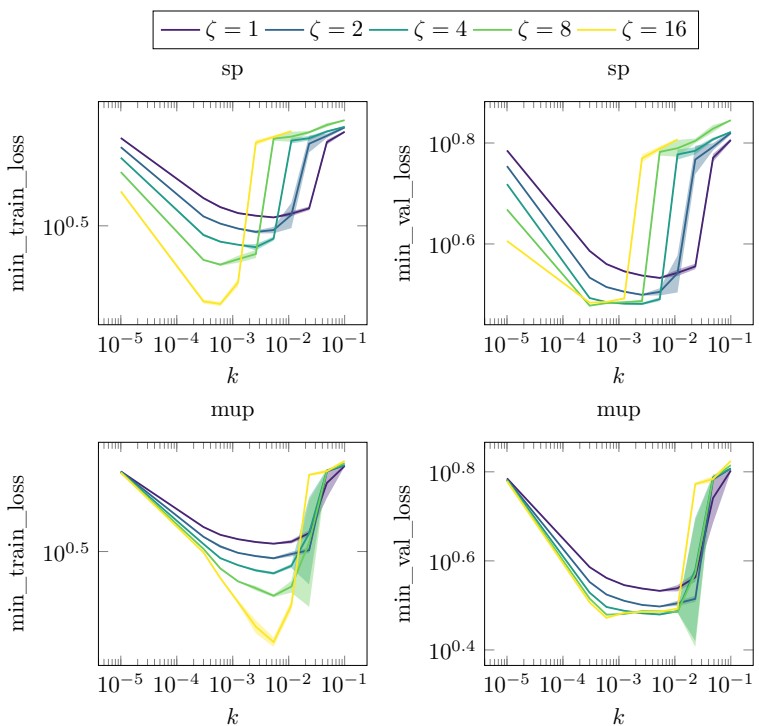

Figure 7: Transformer on WikiText-103. We plot mean (solid) $\pm 1$ std. dev. (shaded) of cross entropy loss versus the learning rate multiplier $k$. These are the cleanest results: we clearly see that the best $k$ stays almost constant under $\mu$P, and that wide networks (mostly) outperform narrow ones.

## 6    Conclusion

This paper is a head-to-head comparison between SP and $\mu$P. We independently reproduced the empirical claims of Yang & Hu (2020) and Yang et al. (2021), while at the same time significantly increasing the scale of the experiments. We confirm that $\mu$P indeed has a number of benefits over SP, even though one might not observe all of them in every setup. In general, $\mu$P stabilizes the optimal learning rate as a function of width and makes wider networks outperform narrower ones. Furthermore, it alleviates divergence issues. However, in terms of both train and validation error, the best $\mu$P network is often worse than the best SP network.

While our results do confirm that $\mu$Transfer is a viable paradigm for hyperparameter optimization, the alternative is not necessarily tuning the hyperparameters of the biggest network directly. In practise, for SP, it is more common to search for hyperparameters by training the biggest network for fewer iterations. It would be interesting to compare that protocol with narrow-to-wide $\mu$Transfer for the same compute budget.

Since $\mu$P is theoretically well-founded and empirically has a consistent stabilizing effect, it merits further investigation. In particular, future research should investigate under what circumstances $\mu$P networks generalize better than their SP counterparts, and whether $\mu$P can be adapted to more consistently outperform SP in terms of generalization.

### Acknowledgments

Calculations were performed at sciCORE (`http://scicore.unibas.ch/`) scientific computing center at University of Basel. We are grateful for the support of Prof. Jiri Cerny of University of Basel.

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
