# OpenReview forum: "A thorough reproduction and evaluation of $\mu$P"
_TMLR — Accepted by TMLR_

### Review · Reviewer_T9E4 · 2024-11-22

**Summary Of Contributions:**

This paper presents an independent empirical reproduction of the claimed benefits of the µP parametrization. The experiments are extensive, ranging from 500 to 0.5B parameters, with evaluations on over 10k neural networks.

**Audience:**

Yes

**Broader Impact Concerns:**

No ethical concerns

**Claims And Evidence:**

Yes

**Requested Changes:**

1. It would be better to report the test accuracy for CIFAR-10 training.
2. Include the base learning rate and explain how it was determined.
3. Conduct large-scale training experiments, e.g., using at least a 1B-parameter model and 1B tokens.
4. It is better to give some practical guidelines for improving the SP/µP training.

**Strengths And Weaknesses:**

Strengths:
1. The question addressed by this work—probing the real utility of µP, given its limited use in published works—is significant.
2. The paper is clearly presented and supported by convincing results.

Limitations:
1. The experiments focus only on small datasets, e.g., CIFAR-10 and the 4000-line Tiny Shakespeare dataset. These may not truly reflect the results on larger-scale datasets, such as those used for LLM pretraining. Additionally, the largest model considered in the paper is relatively small, with only 0.5B parameters.
2. No new methods or insights are provided.

---

> ### Author Response · Authors · 2024-12-18
>
> Thanks for taking the time to review our paper!
>
> * We added two new settings: MLP on MNIST and transformer on WikiText-103. The second one (up to 1B parameters and 1B tokens) is at least 100 times more computationally expensive than the setting in the original paper of Yang et al (up to 350M parameters and 10M tokens). In total, the experiments took about 3200 A100 hours (equivalent to 13000$ on AWS).
> * We now also report more interpretable metrics (e.g. accuracy) for all our settings. The main plots still show loss because this is what muP supposedly stabilizes.
> * Strictly speaking, there is no base learning rate. As seen in Eq. (4), the learning rate is simply k*γ, with k being swept and γ defined in Table 1.
> * We agree that practical guidelines for making muP consistently outperform SP in terms of generalization would be very valuable, which is precisely the point we raise in our conclusion. Nevertheless, there is a lot of value in knowing what to expect for muP as currently implemented, which we think is a significant contribution of our work.

---

### Review · Reviewer_8XW8 · 2024-11-22

**Summary Of Contributions:**

The paper aims to conduct a thorough and head-to-head comparison between the standard parameterization under LeCun initialization of neural networks and the more recently proposed $\mu P$ parameterization, which has been theoretically shown confer several benefits to training but seems to have gained little traction in practice. The authors independently reproduce the empirical claims of the original works that proposed this scheme, scaled up the experiments for more diverse evaluation and presented their findings. Notably, their results indicate that several claims made about optimal learning rates, output norms, gradient norms and weight norms are well-founded and empirically verifiable. However, the authors also present evidence to suggest that these claims do not necessarily lead to better generalization.

**Audience:**

Yes

**Broader Impact Concerns:**

None.

**Claims And Evidence:**

No

**Requested Changes:**

- Typo: I believe the authors intended to write "500M to 0.5B parameters" in the abstract and section 1.2.
- Are the authors able to report evaluation metrics that are more interpretable, for example, accuracies, etc. on the tasks that the models are trained on?
- If the authors can present results on a wider variety of datasets in each setting, their claims about the overall performance differences between the two parameterizations would be better supported.
- What actionable insights can readers and/or practitioners derive from the experiments of this paper?

**Strengths And Weaknesses:**

Strengths:

The goal of thorough and independent empirical evaluation or strong theoretical findings is an important and potentially very impactful one. The paper scales up evaluation of $\mu P$ to networks ranging from 500M to 0.5B parameters, which is a wide enough range to provide insightful information. The authors also make sure to present results that are comparable between $\mu P$ and $SP$. Interpretations and actionable insights from this kind of evaluation would be very valuable to practitioners.

Weaknesses:

- I found the results from this study difficult to interpret. For instance, the authors report training and validation losses in all their settings, to make conclusions about the effective performance in different settings. However, it is difficult to compare and interpret absolute values of losses. The significance of loss values like 0.17 vs. 0.18 (section 4.4) is unclear.
- Since the contribution of this work is an empirical evaluation, similar to a benchmarking study, I find the numbers of architectures (4) and datasets (1 per architecture) to be very low. Evaluating performance differences in such few settings is likely to lead to high-variance conclusions.

---

> ### Author Response · Authors · 2024-12-18
>
> We really appreciate you taking the time to review our paper!
>
> * We now also report more interpretable metrics (e.g. accuracy) for all our settings.
> * In this work we tried to do an extended reproduction, not a benchmarking study. We thus decided to dedicate our compute budget (3200 A100 hours, equivalent 13000$ on AWS) to reproducing the behaviour of muP loss curves as cleanly as possible (with more widths, denser learning rate sweep, more random seeds). In other words, we opted for clean results in a few settings, instead of noisier results in more settings. Nevertheless, we added two new settings: MLP on MNIST and transformer on WikiText-103. The second one (up to 1B parameters and 1B tokens) is at least 100 times more computationally expensive than the setting in the original paper of Yang et al (up to 350M parameters and 10M tokens).
> * The smallest model we tried had five hundred (500) parameters (not 500 million). Our largest model now has 1 billion parameters.
> * We summarize our insights in Sections 1.3 and 5. Our conclusion is that the practical deployment of muP may be premature (since it does not consistently lead to better generalization than SP). Nevertheless, muP shows promise and merits further research.

---

> > ### Comment · Reviewer_8XW8 · 2024-12-18
> > **Response**
> >
> > Thanks for your response!
> >
> > - I don't seem to be able to find the reported results on accuracy - can I request the authors to point out the corresponding tables or figures?
> > - Thanks for your work on adding more datasets, and for clarifying the range of number of model parameters in the experiments.

---

> > > ### Author Response · Authors · 2024-12-18
> > >
> > > * We added the new metrics next to the loss values in the main text (e.g. for accuracy, in 4.2, 4.3 and 4.4). We tried adding them to the table, but then the table becomes too crowded. The main plots still show loss because this is what muP supposedly stabilizes.
> > > * Thanks for you suggestion! The addition of the two new settings definitely made the study more comprehensive.

---

### Review · Reviewer_XHzd · 2024-11-25

**Summary Of Contributions:**

The paper provides an in-depth study on the advantages of Maximal Update Parametrization ($\mu$P) compared to Standard Parametrization (SP). The authors start by providing a background on $\mu$P. They then conduct an extensive set of empirical experiments to identify scenarios where $\mu$P should (or not) be preferred over SP. The findings highlight that $\mu$P offers more stability as the width increases, making it a robust choice for scaling. However, it is also observed that $\mu$P underperforms relative to SP in general settings. Additionally, the study reveals that SP, while effective in several cases, diverges on very wide networks. These insights provide comprehensive comparison on the practical trade-offs between the two methods.

**Audience:**

Yes

**Claims And Evidence:**

Yes

**Requested Changes:**

- The authors could possibly consider using a log scale for all the train/val loss plots to improve readability to better highlight the differences, especially for smaller values.
- Maybe make the figure captions self-contained by providing clear descriptions of what the reader should expect or infer from the figures.
- In Figure 2, some curves are still not clearly visible because of shaded regions. While the color choices show trends, increasing the transparency of the shaded areas could further improve clarity and visibility.

**Strengths And Weaknesses:**

Strengths:

- The writing is overall clear, with a well-defined scope and objective.
- The empirical study is comprehensive and the authors provide clear set of findings, added by details like the number of GPU hours and other relevant experimental setups for reproducibility.
- The findings on generalization and the benefits of $\mu$P, particularly in the context of Transformers, are timely and relevant to current research trends.

Weaknesses:
- While the scope is quite limited, the current version also has some similarities to a technical report in terms of presentation. I think it'd strengthen the paper if authors can highlight/discuss why the observed effects occur in each section Or if such behaviour are indeed theoretically expected. I see answers to these question hinted in only a few cases. I think the authors could provide more thorough discussions, analysis/implications and conclusions from the results as the current paper lacks these.

Questions:
- Section 4.1.2: I'm not sure if one may conclude that SP outperforms $\mu$P in terms of training loss since the observed difference is quite small and curves overlap significantly. Wouldn’t it be better to just interpret these results as $\mu$P exhibiting greater stability in wider networks, rather than claiming that one clearly outperforms other in terms of training loss?
- Section 4.3: I don't understand how the validation loss curve is reminiscent of double descent. The double descent is generally observed across increasing capacity of the network. However, I can't conclude that clearly from Figure 4. Can the authors clarify that?
- One of the findings mentioned $\mu$Transfer enables zero-shot hyperparameter transfer, from narrow networks to wider ones. Have the authors explored $\mu$Transfer experimentally across models/datasets? For example, applying transfer from VGG to ResNet or using ViT on CIFAR-10 as a base setup for training bigger ViT on ImageNet?

---

> ### Author Response · Authors · 2024-12-18
>
> Thanks for your valuable feedback!
>
> * We added more explanations about what is expected throughout the paper (4.2: "Contrary to theory, k shifts to the right for μP", 4.6: "The loss curves are very similar to those in Section 4.5, but exhibit even less noise. This could be explained by the significantly bigger batch size (512 vs 32)" etc.).
> * We now also report more interpretable metrics (e.g. accuracy) for all our settings, which actually show a non-negligible difference.
> * The "double descent" was actually a misinterpretation on our part. We removed the wrong explanation. Moreover, rerunning the experiments produces virtually the same peak (we added a corresponding footnote).
> * We mainly focused on evaluating the main claims of the original paper of Yang et al, and hence did not investigate transfer across architectures or datasets. Nevertheless, by comparing Figure 4 and 5 (VGG and ViT on CIFAR-10) and Figure 6 and 7 (transformer on Tiny Shakespeare and WikiText-103) we see that neither holds in practice, since the optimal learning rate multipliers are quite different.
> * Thank you for the log scale suggestion. We now employ it in the latest revision.
> * We added concise descriptions to the figures to make them self-contained.
> * We changed the color scheme, made the lines thicker and reduced the opacity of the shaded areas.

---

### Decision · Action_Editor_pNtC · 2024-12-29

**Recommendation:** Accept with minor revision

**Comment:**

The paper presents an empirical study comparing Standard Parametrization (SP) and μP across settings. Reviewer recommendations are mixed (1 leaning accept, 2 leaning reject); for some reviewers the authors have NOT addressed many key concerns through their revision:

Strengths:

- Clear experimental methodology and comprehensive evaluation within scope
- Significant computational investment (3200 A100 hours)
- Added interpretable metrics (accuracy) and new experimental settings
- Tempered claims to match empirical evidence
- Important practical insights about μP's benefits and limitations

Remaining Concerns:

- Some performance differences are small and hard to interpret definitively
- Some theoretical explanations for observed behaviors could be strengthened

Given TMLR's criteria emphasizing thorough empirical evaluation and reproducibility, and considering the authors' careful revisions and responses, I believe this paper makes a valuable contribution despite its limitations. The work provides important practical insights about μP's applicability while maintaining appropriate scope and claims supported by evidence.

Recommendation: Accept with minor revision to address remaining clarity issues in result interpretation and theoretical discussion.

**Audience:**

Yes, this work would be of interest to TMLR's audience for several reasons:

- It provides empirical investigation of an important theoretical advancement (μP) that has seen limited practical adoption
- The findings are relevant to practitioners working on neural network training and scaling
- The reproduction study helps bridge the gap between theoretical promises and practical implementation
- The results provide insights about when μP may or may not be advantageous over SP
- The work raises important questions about generalization behavior that would interest both theoretical and applied researchers

**Claims And Evidence:**

The paper's claims about the empirical comparison between Standard Parametrization (SP) and μP are mostly supported by evidence, though with some limitations:

- The experimental results are thorough within their scope, covering networks from 500 parameters to 1B parameters
- The authors added additional metrics (accuracy) and settings (MLP on MNIST, transformer on WikiText-103) in response to reviewer concerns
- The computational investment is significant (3200 A100 hours) and focused on clean reproduction rather than broad benchmarking
- Some results show small differences that could be attributed to randomness, but the authors have appropriately tempered their claims to match the evidence

However, there are some limitations in the evidence:

- Relatively small number of architectures (4) and datasets
- Some results show very close performance between SP and μP, making interpretation challenging

**Resubmission Of Major Revision:**

The authors may consider submitting a major revision at a later time.

---

> ### Author Response · Authors · 2025-01-23
> **Camera-ready version**
>
> We thank both the AC and all the reviewers for all their comments! Thanks to them, our paper has improved considerably since the first version. We just uploaded the camera-ready version. In the latest revision:
> * We added the confidence intervals in Table 2, adjusted even more the wording in result interpretation so that it reflects the differences observed and added a comment on the caption of the table about the results' variance (specifically, that, in general, the variance is small, making our results trustworthy).
> * Improved the theoretical discussions (second paragraph of Sec. 1.1, second paragraph of Sec. 4.1.1, Sec. 4.4, Sec. 5), so that it is more clear what should or should not be expected.
> * Improved wording even more
> * Deanonymized the document (+added code repo)